spectroscopy

terahertz spectroscopy, food additive, machine learning, quantitative analysis

**Author for correspondence:**
Xudong Sun
e-mail: 874916937@qq.com

This article has been edited by the Royal Society of Chemistry, including the commissioning, peer review process and editorial aspects up to the point of acceptance.

# Generalized regression neural network association with terahertz spectroscopy for quantitative analysis of benzoic acid additive in wheat flour

Xudong Sun, Junbin Liu, Ke Zhu, Jun Hu, Xiaogang Jiang and Yande Liu

School of Mechatronics and Vehicle Engineering, East China Jiaotong University, Nanchang 330013, People's Republic of China

XS, 0000-0001-6743-1348

Investigations were initiated to develop terahertz (THz) techniques associated with machine learning methods of generalized regression neural network (GRNN) and back-propagation neural network (BPNN) to rapidly measure benzoic acid (BA) content in wheat flour. The absorption coefficient exhibited a maximum absorption peak at 1.94 THz, which generally increased with the content of BA additive. THz spectra were transformed into orthogonal principal component analysis (PCA) scores as the input vectors of GRNN and BPNN models. The best GRNN model was achieved with three PCA scores and *spread* value of 0.2. Compared with the BPNN model, GRNN model to powder samples could be considered very successful for quality control of wheat flour with a correlation coefficient of prediction ($r_p$) of 0.85 and root mean square error of prediction of 0.10%. The results suggest that THz technique association with GRNN has a significant potential to quantitatively analyse BA additive in wheat flour.

## 1. Introduction

Benzoic acid (BA), a food additive, is widely used to preserve different kinds of foods. The United States Food and Drug Administration, however, considers BA approved for use as food additives safe for humans when consumed in small amounts. But the long-term consumption of wheat flour containing BA

additive may cause the accumulation of BA in the liver, resulting in cumulative poisoning. In recent years, debate on the addition of BA to wheat flour has increased in China. Hence, it is necessary to develop a suitable and rapid method for product supervision and sampling inspection to determine BA in wheat flour.

With the current method for analysis of food additives, BA is commonly measured accurately by gas chromatography (GC) or high-performance liquid chromatography (HPLC) [1]. But these methods could not be used for on-site detection because of time consumption and requirement of chemical reagents. Terahertz (THz) wave, located between radio wave and infrared light, possesses the characteristics of the fingerprint spectrum and low-density transmission. Therefore, THz spectroscopy is able to penetrate the food substrate to obtain THz spectra of various components. Many biological molecules have unique spectral fingerprints in the THz frequency range, which means that THz spectroscopy can be used to identify them. THz spectroscopy as a powerful tool for non-destructive detection technology is increasingly applied in the fields of agriculture and food industry [2,3]. A typical THz system of time-domain spectroscopy (TDS) has been developed to evaluate the optical properties of biological molecules [4]. The absorption spectrum and refractive index of BA were measured by THz-TDS technology in the range of 0.5–4 THz [5,6]. For the past few years, THz-TDS technology together with chemometrics algorithms has been attempted to quantitatively detect, mainly because of the advantages of low-cost, non-destructive and generally simple sample pretreatment [7–9]. Among these algorithms, partial least square (PLS) regression was the most widely used algorithm for exploring the linear relationship between THz spectra and target references, for example, quantitative analysis of tetracycline hydrochloride and amino acid [10–12]. However, nonlinear relations between THz spectra and target component may appear because of variation of the sample, change of environmental condition and instrumental noise. Artificial neural networks (ANN) have the potential to deal with the nonlinear problem in the THz spectra analysis [13,14]. However, it should be pointed out that general regression neural networks (GRNN) were adopted instead of frequently employed back-propagation neural networks (BPNN), because GRNN has the following advantages: single-pass learning so no back propagation is required, high accuracy in the estimation for uses of Gaussian function, and it can handle noises in the inputs [15,16]. BPNN requires selecting training parameters and defining network architectures in the process of training. But for GRNN, the only weight to be learned is the smoothing factor $\sigma$, the width of radial basis functions. Because of its good performance, the GRNN has been extensively applied in rainfall-runoff modelling and in intermittent flow estimation [17–20]. These features are favourable to efficiently investigating the relationship between THz spectra and BA concentrations.

# 2. Material and methods

## 2.1. Samples preparation

The BA powder was purchased from Sigma-Aldrich Corporation and used without further purification. The wheat flour was purchased from a local supermarket and had been validated without BA powder by HPLC analysis. The BA powder samples were crushed into small particles that were sufficiently smaller than the THz wavelength to reduce baseline offsets at higher frequencies. These partials were mixed carefully with wheat flour at several different concentrations (from 0.08% to 1.14%, g/100 g), and three replicates were prepared for each concentration. Then the mixture was compressed into pellets with the diameter of 13 mm under pressure of 10 MPa by use of tablet press. The mechanically determined pellet thickness ranged from 1 to 2 mm to provide a sufficient path length to eliminate the effect of the multiple reflections that occurred between the two surfaces of the pellet sample in the spectra. In addition, 10 real samples were collected from a local oil and food testing institution. These samples were made into tablets according to the same above procedures as the external testing ones.

## 2.2. THz measurement

The absorption spectra were recorded with the TAS7500SU THz-TDS system working in transmission mode, provided by Advantest Corporation. The detail of this system can be found in the literature [2]. The system includes two ultra-short pulse fibre lasers, which are ensured to synchronized control. The central wavelength and maximum output power of these pulses are 1550 nm and 50 mW, respectively.

These pulses provide an extremely short pulse width less than 50 fs and low jitter below 50 fs. The system achieves the sampling rate with 8 ms per scan and ultra-wide frequency band extending to 7 THz. The experiment was carried out at room temperature, under the circumstance of a dry-air purged container with the relative humidity of 0%. Three measurements were recorded for each sample to reduce the random error. The reference waveform was collected when the THz pulses passed through a sample holder without sample mounted in it.

## 2.3. Parameters extraction

A fast Fourier transform (FFT) was adopted to acquire the spectral distribution of the THz pulse in the frequency. The sample's absorption coefficient ($\alpha$) could be calculated with the below equations:

$$\alpha(\omega) = \frac{2}{d} \ln \left( \frac{4n(\omega)}{\rho(w)(n(w)+1)^2} \right) \tag{2.1}$$

and

$$n(\omega) = 1 + \frac{\phi(w)c}{\omega d}, \tag{2.2}$$

where $c$, $\omega$ and $d$, are the light speed in vacuum, the frequency and the sample's thickness, respectively. The $\rho(\omega)$ and $\Phi(\omega)$ represent, respectively, the amplitude ratio and phase difference between the reference and sample.

## 2.4. GRNN algorithm

Differing from BPNN, GRNN is a variation to radial basis neural networks and consists of four layers: input, pattern, summation and output layers [21–23]. Terahertz spectra are used as the input vectors in the first layer. The second layer has the pattern units, and the outputs of this layer are passed onto the summation units in the third layer. The final layer covers the output units. Its architecture is shown in figure 1. GRNN replaces the sigmoid activation function often used in ANN with a radial basis function (RBF) and achieves the estimation of the probability density function using Parzen's non-parametric estimator [15]. The predicted value is simply a weighted average of the target values of training patterns close to the given input pattern. The smoothing factor $\sigma$, representing the width of RBF, is the only adjustable parameter. The details of GRNN are as follows:

$$Y(x) = \frac{\sum_{k=1}^{N} y_k K(x, x_k)}{\sum_{k=1}^{N} K(x, x_k)} \tag{2.3}$$

and

$$K(x, x_k) = e^{-(x-x_k)^T (x-x_k)/2\sigma^2}, \tag{2.4}$$

where $Y(x)$ is the prediction value of input $x$, $y_k$ is the activation weight for the pattern layer neuron at $k$, and $K(x, x_k)$ is the RBF as formulated above.

# 3. Results and discussion

## 3.1. Statistics of measured BA concentration

The 160 samples were divided into training and validation datasets according to an approximate scale of $3:1$. The BPNN or GRNN model is initially fit on the training dataset to fit the parameters e.g. weights of connections between neurons. The validation dataset follows the same probability distribution as the training dataset. The validation dataset provides an unbiased evaluation of a model fit on the training dataset while tuning hyper-parameters, e.g. the number of hidden units in a neural network. And the test dataset is used to provide an unbiased evaluation of the final BPNN or GRNN model fit on the training dataset. The external dataset was provided by a local grain and food testing institution. It was used as an independent dataset that has never been used in training. The distribution and statistical results of BA concentration values are shown in figure 2 and table 1, respectively.

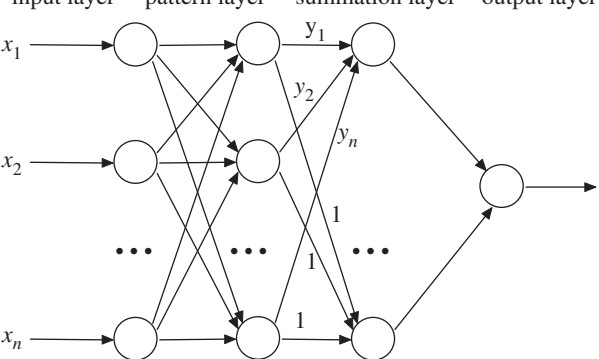

input layer    pattern layer    summation layer    output layer

**Figure 1.** The architecture of the GRNN.

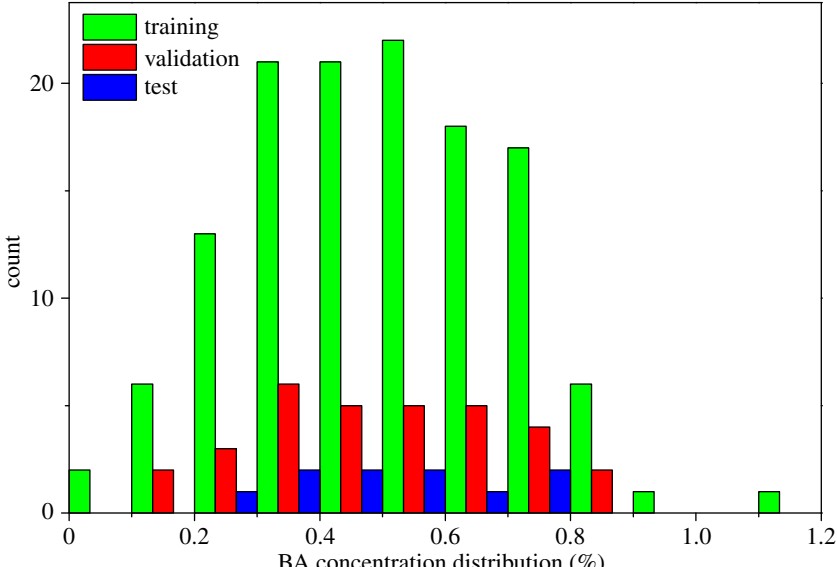

**Figure 2.** The histograms of calibration, validation and test sets.

## 3.2. Analysis of spectral characteristic

First Savitzky–Golay derivative with window width of 9 and polynomial order of 2 combinations with smoothing average was adopted to remove the baseline shift and amplify absorption peak. The first derivative absorbance coefficients of mixture, wheat flour and BA samples in the 1.6–2.8 THz frequency region were shown in figure 3. The regions below 1.6 THz or beyond 2.8 THz were considered as ineffective data, because of a relatively low signal-to-noise ratio (SNR). The maximum absorption peak at 1.94 THz arises from the rocking of carboxyl and phenyl out of the molecular plane, which belongs to intra-molecular vibration [5]. But the absorption coefficient plot of wheat flour does not present an obvious peak at the regions of 1.6–2.8 THz. The absorption coefficient generally increased as the content of BA increased. Therefore, the line-fitted equation was established for investigating the relationship between the peak of 1.94 THz and BA concentrations, the results were shown in figure 4. The fitted model was assessed by correlation coefficient ($r$), root mean square error of prediction (RMSEP) and limit of detection (LOD). A better model should obtain higher $r$, and lower RMSEP and LOD values. According to this principle, further investigation should be executed to mine THz spectra for accuracy improvement. For Lambert–Beer's law, only the fingerprint peak can give a better relationship equation using the pretreatment liquid sample. In this case, THz spectroscopy cannot obtain a high correlation coefficient because the sample is relatively complex and only fingerprint peak cannot give enough information. Therefore, it requires more variables as the input vectors to improve the performance of the models. The LOD with 99.86% confidence interval

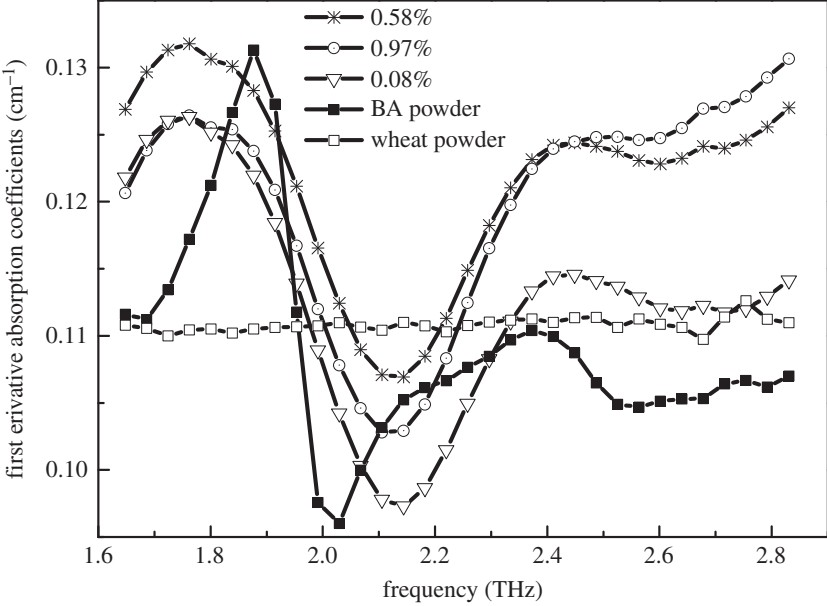

**Figure 3.** Absorbance spectra of the mixture, wheat flour and BA samples in the 1.6–2.8 THz frequency region.

**Table 1.** Statistics of calibration and prediction sets of BA concentration in wheat flour. N, number of samples; s.d., standard deviation; CV, coefficient of variation.

| dataset | N | range (%) | mean (%) | s.d. (%) | CV (%) |
|---|---|---|---|---|---|
| training | 128 | 0.08–1.14 | 0.50 | 0.20 | 40.00 |
| validation | 32 | 0.12–0.90 | 0.50 | 0.20 | 40.00 |
| test | 10 | 0.23–0.80 | 0.51 | 0.18 | 35.29 |

can be calculated from the fitting curve based on significant peaks in THz absorption coefficient in formula (3.1) [24].

$$LOD = \frac{3\sigma}{m},$$

(3.1)

where $\sigma$ is the standard error of predicted concentration, and $m$ is the slope of the fitting curve. In a fitting model, $\sigma$ equals RMSEP.

## 3.3. Development of GRNN model

The number of input vectors will influence the architecture and performance of the GRNN model. Generally, the minimal architecture of GRNN makes it easy to obtain a better generalization of data relation. Principal component analysis (PCA) is a bilinear modelling method which gives an interpretable overview of the main information in a multidimensional data table. The information carried by the original variables is projected onto a smaller number of underlying variables called principal components. The first principal component covers as much of the variation in the data as possible. The second principal component is orthogonal to the first and covers as much of the remaining variation as possible, and so on. THz spectrum includes 200 spectral variables in a range of 1.6–2.8 THz, and some variables may contain irrelevant information for regression. Therefore, PCA was applied to transform original THz spectra into a new axis and obtain the PCA scores as new variables. The scores show the locations of the samples along each model component and can be used to detect sample patterns, groupings, similarities or differences. The score plots of first and second principal components were shown in figure 5. The distribution of the calibration samples covered the validation and test ones.

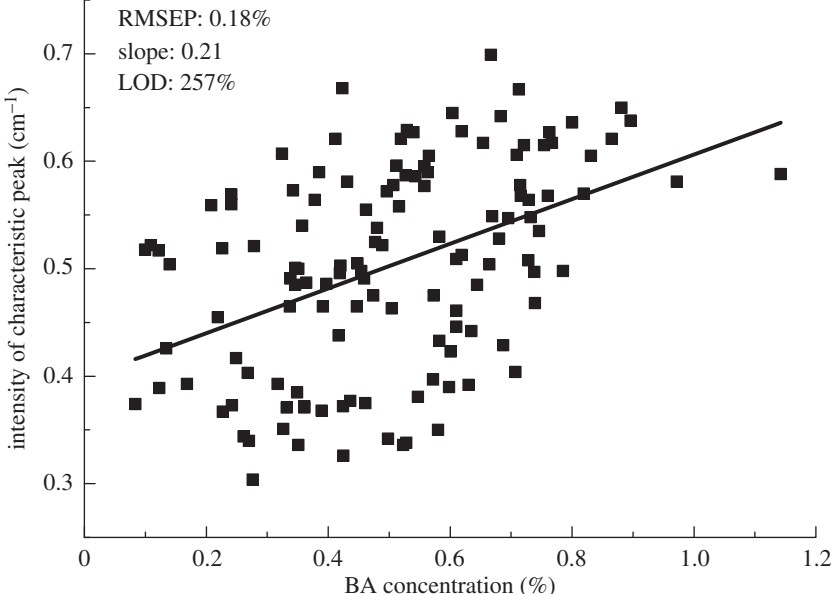

**Figure 4.** Fitting model between characteristic peak and BA concentrations.

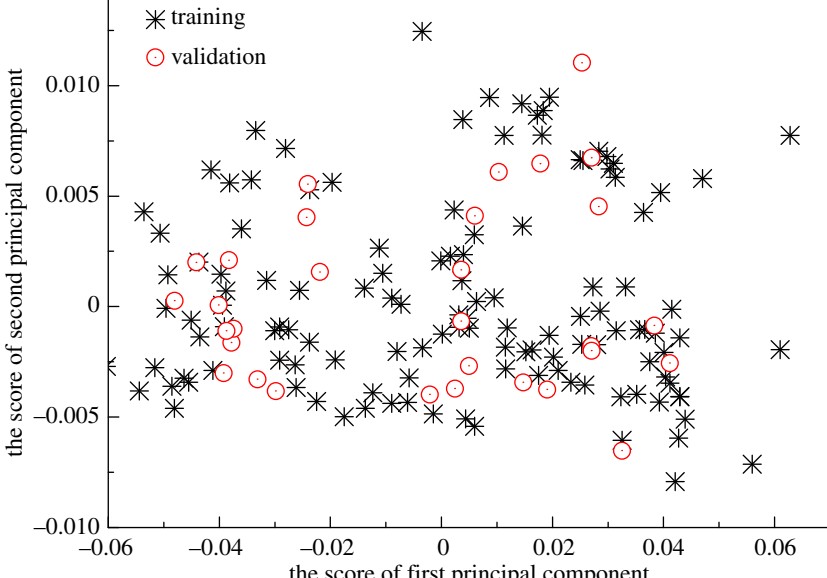

**Figure 5.** The score plots of first and second principal components.

The variance is computed as the mean square of deviations from the mean. It is equal to the square of the standard deviation. The PCA scores accounted for the greatest amount of variability and varied from 97.12% to 99.99% (figure 6), which presented in the THz spectra collected using THz-TDS system.

GRNN is a highly parallel radial basis network model generated by the function *newgrnn* in Matlab software. The number of input vectors and smooth factor ($\sigma$) of RBF are two important parameters that influenced the performance of GRNN model. The principal component scores were chosen as the input vectors of GRNN model. The number of principal components varied from 1 to 10. The smooth factor behaves as a regularization parameter of RBF. When the smoothing parameter $\sigma$ is made large, the estimated density is forced to be smooth and in the limit becomes a multivariate Gaussian with covariance $\sigma^2$. On the other hand, a smaller value of $\sigma$ allows the estimated density to assume non-Gaussian shapes, but with the hazard that wild points may have too great an effect on the estimate [25]. In this case, the method of circle training was adopted to optimize the *spread* in the range of 0.01–2, and the interval was 0.05. The training and validation datasets were used to create GRNN model and optimize the parameters; the results are shown in figure 7. The optimal GRNN model was obtained with three principal components and smooth factor of 0.2.

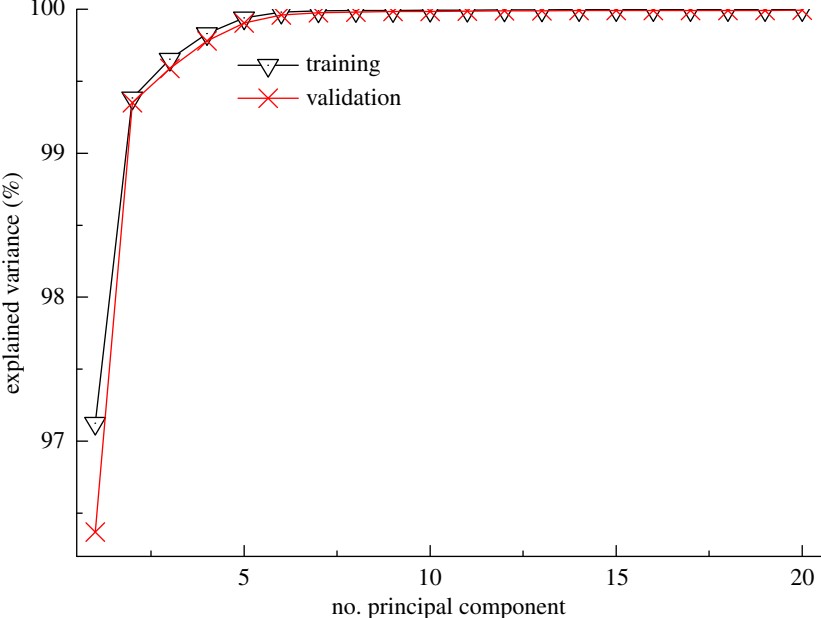

**Figure 6.** Variance plots for different principal components.

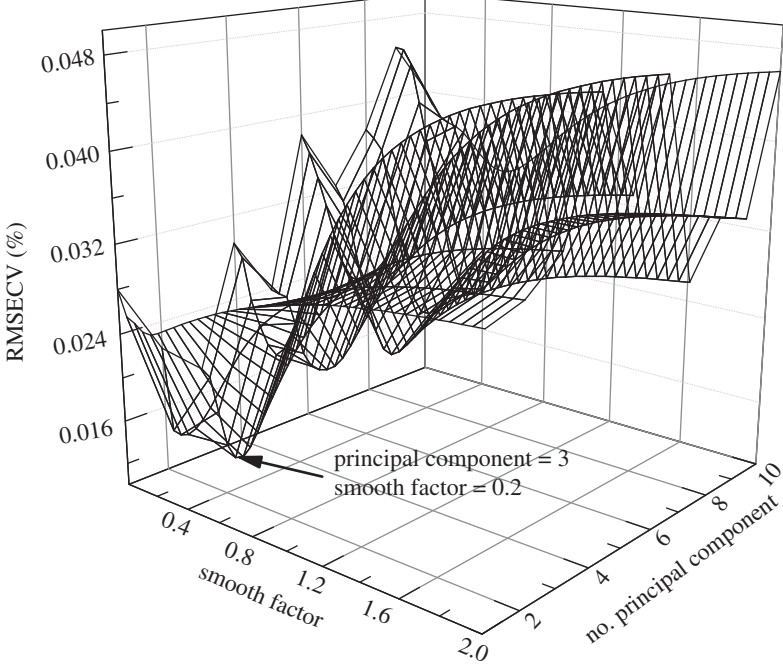

**Figure 7.** Optimization for number of principal components and smooth factor.

## 3.4. Comparison of GRNN and BPNN models

To compare with GRNN model, BPNN was developed with three layers of neurons (input, hidden and output). Each node in the input and hidden layers is connected to each of the nodes in the next layer with a weighting factor associated with it. These weights are modified using the back-propagation algorithm during the training process. The transfer functions of hidden and output layers were *tansig* and *purelin* functions, respectively. The training and performance functions were *trainlm* and mean square error (MSE), respectively. The goal error was set as 0.001. The time of training was set as 200. The early stopping is a default method to avoid BPNN networks overfitting. The available data have been divided into the training and validation sets. The former is used for computing the gradient and updating the network weights and biases. The latter is monitored during the training process. The

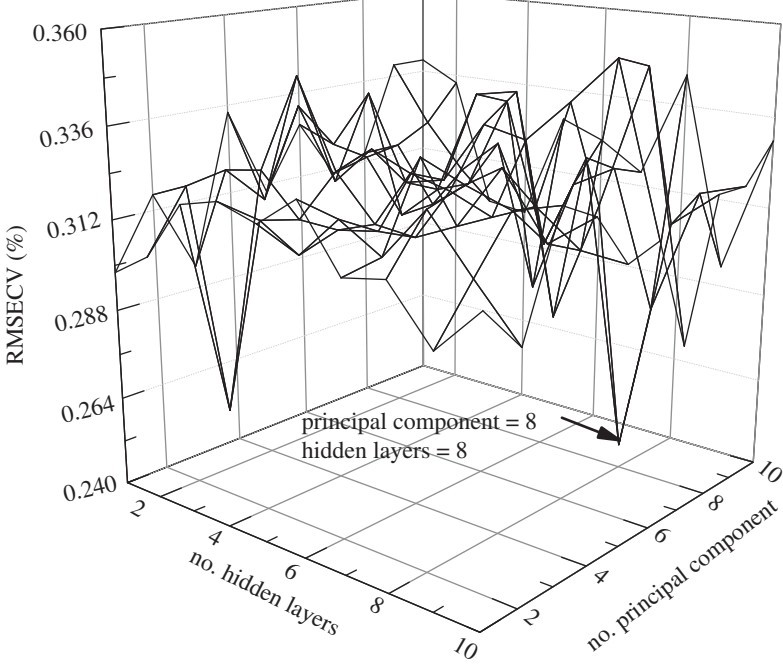

**Figure 8.** The optimized results of BPNN for number of input vectors and hidden layers.

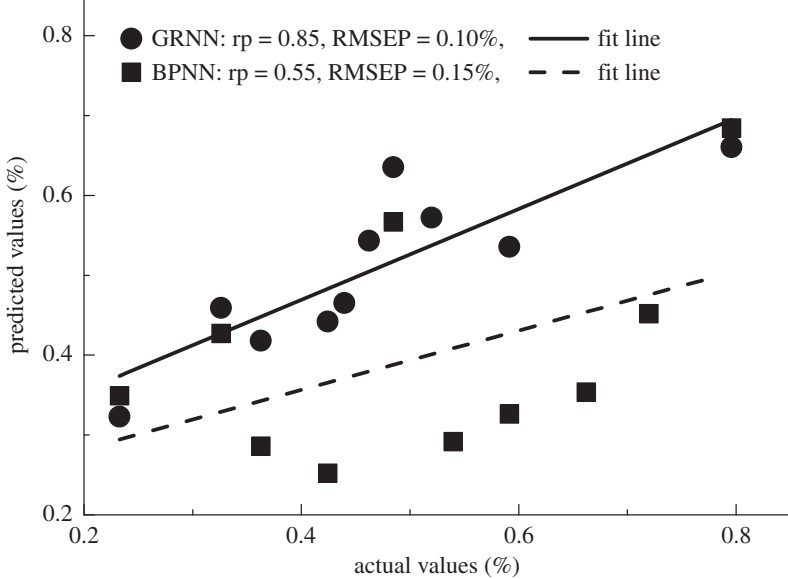

**Figure 9.** Comparison of predictive abilities for GRNN and BPNN models.

validation error normally decreases during the initial phase of training, as does the training set error. However, when the network begins to overfit the data, the error on the validation set typically begins to rise. When the validation error increases for a specified number of iterations (net.trainParam.max_fail), the training is stopped, and the weights and biases at the minimum of the validation error are returned. The number of input vectors and hidden layers were optimized, and the results are shown in figure 8. The training and validation datasets were applied to create and optimize the model. According to the best model should have the lowest root MSE of cross-validation (RMSECV), the best BPNN model was achieved with eight principal components and eight neurons of the hidden layer.

The practical predictive abilities of the best GRNN and BPNN models were subsequently evaluated with an independent dataset, which was provided from a local oil and food test institution. Compared with the BPNN model, GRNN model obtained the highest predictive accuracy with a correlation coefficient of prediction (*r*) of 0.85 and RMSEP of 0.10%. The results are shown in figure 9. It implied

that the BA concentration values of mixture samples could be determined by sophisticated machine learning methods of GRNN. Based on the discussion above, GRNN was more suitable than BPNN for BA concentration determination, because GRNN actually obtained a good balance between accuracy and speed. The running time of GRNN was 0.12 s. Hence, it has more potential for real-time applications with a comparable accuracy.

The paired *t*-test method was applied to evaluate the difference between HPLC measurement and THz analysis for the external dataset. The calculated *t* values were 0.05 for BA content, while *t* value was lower than *t*(0.05, 18) of 0.26. It could be seen from paired *t*-test results that the THz spectroscopy combined with GRNN method and reference methods did not show a significant difference for BA content.

# 4. Conclusion

The overall results sufficiently demonstrate that BA concentration values in wheat flour could be determined by THz spectroscopy associated with machine learning method of GRNN. The number of input vectors and spread of RBF were optimized for improving the predictive abilities of GRNN model. And the best GRNN model was obtained with three PCA scores of input vectors and *spread* value of 0.20. Compared with BPNN model, GRNN model to powder samples could be considered very successful for quality control of wheat flour with $r_\mathrm{p}$ of 0.85 and RMSEP of 0.10%. Moreover, BA powder exhibited a maximum absorption peak at 1.94 THz. These results suggest that THz technique in association with GRNN may have commercial and regulatory potential to avoid time-consuming work, and costly and laborious chemical analysis for BA additive in wheat flour.

Data accessibility. Data of THz absorption coefficients and BA contents are named as data.mat. In this file, the X is the first derivative spectra of THz absorption coefficients, the Y is BA contents and the fz is the frequency. The DemoGRNN.m is the program of GRNN algorithm. DemoBPNN is the program of BPNN for compared with GRNN. The rms.m is for root mean square of error. BP.m is the source code of BPNN. The data are available from the Dryad Digital Repository at https://datadryad.org/resource/doi:10.5061/dryad.945c410 [26].

Authors' contributions. X.S. analysed the results and drafted the manuscript. J.L. collected the samples, recorded the spectra and processed the spectra tighter with X.S. K.Z. participated in all the tests. J.H. helped with interpretation of the data. X.J. revised the manuscript. Y.L. reviewed the draft.

Competing interests. We declare we have no competing conflicts.

Funding. Funding support from Outstanding Youth Talent Program of Jiangxi Province (20171BCB23060), Education Department Project of Jiangxi Province (GJJ160478), China Scholarship Council (201808360317), Jiangxi Association for Science and Technology (JAST) and Doctor Start-up Project (368).

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
