## [Reviewer comments · Royal Society Open Science]

Review History

RSOS-180765.R0 (Original submission)

Review form: Reviewer 1

Is the manuscript scientifically sound in its present form?

No

Are the interpretations and conclusions justified by the results?

No

Is the language acceptable?

Yes

Is it clear how to access all supporting data?

Yes

Do you have any ethical concerns with this paper?

No

Have you any concerns about statistical analyses in this paper?

Yes

Recommendation?

Reject

Comments to the Author(s)

The research seems interesting, but this paper is of low scientific quality, hence I can not support its publication. Some important information is missing, references are also needed, there is no discussion – pros and cons of this approach are not highlighted. Moreover, it seem that some methodological flows exist. Its technical soundness must be improved. etc

If we start from the Introduction, my suggestion are as follows:

“The data available from THz measurements is generally never enough for BPNN or LS-SVM” – Unclear.

References are needed for the statements made on PNN and GRNN in Introduction.

“GRNN can be used for regression, prediction and classification...” – The difference between reg. and pred. is? Moreover, for classification is used GRNN counterpart, i.e. PNN.

“...approximately normal around the averaged value of 10.25%.” - Histogram with appropriate test will be useful here.

“for modelling applicable model.” – You mean creation of?

“line fitted equation was established for investigating the relationship” – But no discussion is provided on this.

PCA related results must be presented.

“The different PCA scores were selected as the input vector of GRNN model for investigating the influence of different input dimensionalities, the results were shown in figure 4.” – You have actually change the number of PCs?

Fig 4 data - Is this obtained on test set? If so, this is not good practice, because the credibility of test set is based on principle that it can not be used for model parameter determination. You should use eugenvalues as criterion for PC relevancy.

“The larger the spread, the smoother the function approximation will be. Oppositely, the smaller the spread, the stronger the approximation to the sample will be.” - This is not actually the case, and what the stronger approximation means?

“In this case, the method of circle training” – Details are needed.

In section 3.3. the GRNN results are not presented.

Fig 5. – What series 3 to 10 means?

“The best BPNN model was achieved with seven input vectors of PCA scores and six neurons of hidden layer.” - You can not compare models with different inputs!

Fig 6. - What dataset was used to determine those meta-parameters?

And finally, GRNN results from figure 7 should be presented in section 3.3. In fig 7 keep only BPNN results. Also, add table with performance metrics for those two models.

To conclude, the key flaw is that a third dataset is missing. You can not create a reliable ANN without three dataset: training for weights determination, validation for meta-parameters (e.g. number of inputs or spread in GRNN and similar) determination, and test set to assess final model. If you make multiple training runs until you get something that works best on the test data, you have just rendered the test set as training one, and the model you chose as “the best “ will has to be tested once again on “unseen” data, because you've essentially created a ANN model specifically for the test set.

Review form: Reviewer 2

Is the manuscript scientifically sound in its present form?

No

Are the interpretations and conclusions justified by the results?

No

Is the language acceptable?

Yes

Is it clear how to access all supporting data?

Not Applicable

Do you have any ethical concerns with this paper?

No

Have you any concerns about statistical analyses in this paper?

No

Recommendation?

Major revision is needed (please make suggestions in comments)

Comments to the Author(s)

See attached files for comments (Appendix A).

Review form: Reviewer 3

Is the manuscript scientifically sound in its present form?

No

Are the interpretations and conclusions justified by the results?

No

Is the language acceptable?

Yes

Is it clear how to access all supporting data?

Not Applicable

Do you have any ethical concerns with this paper?

No

Have you any concerns about statistical analyses in this paper?

Yes

Recommendation?

Reject

Comments to the Author(s)

This paper describes a method for the estimation of benzoic acid in wheat flour based on an artificial neural networks (ANN) regression of terahertz spectroscopy data. This paper has some critical problems and the most important are:

(i) The LOD (0.60%) of the methodology is markedly above the normal percentage values for BA in wheat flour (0.05 to 0.1%).

(ii) The prediction set is indeed a test set. Several unknown samples must be assayed and the results compared with those obtained with a standard method.

(iii) Fig. 3 must be discussed. Indeed, the response seems random and do show any observable trend.

(iv) Why PCA was required as data pre-processing. What are the composition and shape of the six components? Why PC Regression was not assessed or PLS?

(v) In the case of the ANN, how many iterations were used and what criteria were used to avoid overfitting.

Decision letter (RSOS-180765.R0)

31-Jul-2018

Dear Dr Sun:

Manuscript ID: RSOS-180765

Title: "Generalized regression neural network association with terahertz spectroscopy for quantitative analysis of benzoic acid additive in wheat flour"

Thank you for submitting the above manuscript to Royal Society Open Science. Your paper was sent to reviewers and their comments are included at the bottom of this letter.

In view of the concerns raised by the reviewers, the manuscript has been rejected in its current form. However, a new manuscript may be submitted which takes into consideration these comments.

Please note that resubmitting your manuscript does not guarantee eventual acceptance, and that your resubmission will be subject to peer review before a decision is made.

Your resubmitted manuscript should be submitted by 28-Jan-2019. If you are unable to submit by this date please contact the Editorial Office.

Yours sincerely,
 Dr Laura Smith, MRSC
 Publishing Editor, Journals
 Royal Society of Chemistry,
 Thomas Graham House,
 Science Park, Milton Road,
 Cambridge, CB4 0WF, UK

Royal Society Open Science - Chemistry Editorial Office

On behalf of the Subject Editor Professor Anthony Stace and the Associate Editor Dr Hazel Cox

REVIEWER(S) REPORTS:

Associate Editor Comments to Author ():

RSC Associate Editor:

Comments to the Author:

(There are no comments.)

RSC Subject Editor:

Comments to the Author:

(There are no comments.)

Reviewers' Comments to Author:

Reviewer: 1

Comments to the Author(s)

The research seems interesting, but this paper is of low scientific quality, hence I can not support its publication. Some important information is missing, references are also needed, there is no discussion – pros and cons of this approach are not highlighted. Moreover, it seem that some methodological flows exist. Its technical soundness must be improved. etc

If we start from the Introduction, my suggestion are as follows:

“The data available from THz measurements is generally never enough for BPNN or LS-SVM” – Unclear.

References are needed for the statements made on PNN and GRNN in Introduction.

“GRNN can be used for regression, prediction and classification...” – The difference between reg. and pred. is? Moreover, for classification is used GRNN counterpart, i.e. PNN.

“...approximately normal around the averaged value of 10.25%.” - Histogram with appropriate test will be useful here.

“for modelling applicable model.” – You mean creation of?

“line fitted equation was established for investigating the relationship” – But no discussion is provided on this.

PCA related results must be presented.

“The different PCA scores were selected as the input vector of GRNN model for investigating the influence of different input dimensionalities, the results were shown in figure 4.” – You have actually change the number of PCs?

Fig 4 data - Is this obtained on test set? If so, this is not good practice, because the credibility of test set is based on principle that it can not be used for model parameter determination. You should use eugenvalues as criterion for PC relevancy.

“The larger the spread, the smoother the function approximation will be. Oppositely, the smaller the spread, the stronger the approximation to the sample will be.” - This is not actually the case, and what the stronger approximation means?

“In this case, the method of circle training” – Details are needed.

In section 3.3. the GRNN results are not presented.

Fig 5. – What series 3 to 10 means?

“The best BPNN model was achieved with seven input vectors of PCA scores and six neurons of hidden layer.” - You can not compare models with different inputs!

Fig 6. - What dataset was used to determine those meta-parameters?

And finally, GRNN results from figure 7 should be presented in section 3.3. In fig 7 keep only BPNN results. Also, add table with performance metrics for those two models.

To conclude, the key flaw is that a third dataset is missing. You can not create a reliable ANN without three dataset: training for weights determination, validation for meta-parameters (e.g. number of inputs or spread in GRNN and similar) determination, and test set to assess final model. If you make multiple training runs until you get something that works best on the test data, you have just rendered the test set as training one, and the model you chose as “the best” will has to be tested once again on “unseen” data, because you've essentially created a ANN model specifically for the test set.

Reviewer: 2

Comments to the Author(s)
See attached files for comments

Reviewer: 3

Comments to the Author(s)

This paper describes a method for the estimation of benzoic acid in wheat flour based on an artificial neural networks (ANN) regression of terahertz spectroscopy data. This paper has some critical problems and the most important are:

(i) The LOD (0.60%) of the methodology is markedly above the normal percentage values for BA in wheat flour (0.05 to 0.1%).

(ii) The prediction set is indeed a test set. Several unknown samples must be assayed and the results compared with those obtained with a standard method.

(iii) Fig. 3 must be discussed. Indeed, the response seems random and do show any observable trend.

(iv) Why PCA was required as data pre-processing. What are the composition and shape of the six components? Why PC Regression was not assessed or PLS?

(v) In the case of the ANN, how many iterations were used and what criteria were used to avoid overfitting.

Author's Response to Decision Letter for (RSOS-180765.R0)

See Appendix B.

RSOS-190485.R0

Review form: Reviewer 2

Is the manuscript scientifically sound in its present form?

Yes

Are the interpretations and conclusions justified by the results?

Yes

Is the language acceptable?

Yes

Is it clear how to access all supporting data?

Yes

Do you have any ethical concerns with this paper?

No

Have you any concerns about statistical analyses in this paper?

No

Recommendation?

Accept as is

Comments to the Author(s)

The manuscript is substantially improved and the authors have reply to the all comments. I appreciate the work done by the authors to improve the manuscript. The paper is acceptable and no further revision is needed.

Review form: Reviewer 3

Is the manuscript scientifically sound in its present form?

No

Are the interpretations and conclusions justified by the results?

No

Is the language acceptable?

No

Is it clear how to access all supporting data?

No

Do you have any ethical concerns with this paper?

No

Have you any concerns about statistical analyses in this paper?

Yes

Recommendation?

Reject

Comments to the Author(s)

This paper describes the analysis of benzoic acid (BA) in wheat flour using terahertz spectroscopy (TS) and principal component analysis (PCA) and artificial neural networks (ANN). Although the subject of this paper may be useful, the work here presented is quite insufficient. The following are the main critical points:

- Is the percentage of BA realistic in real applications? Indeed, up to 20% is a huge amount of BA! Although in the paper, some figures show a concentration of BA up to 0,20%. The maximum concentration was 20 or 0,20%?
- ANN works as a black box that needs optimization. For example, the number of layers, etc. But also, the number of iterations needs to be monitored to avoid over-fitting. What optimization strategy was used?
- The analytical methodology must be validated by the analysis of real samples and the results must be compared with other results obtained by standard methodologies.

Decision letter (RSOS-190485.R0)

24-Apr-2019

Dear Dr Sun:

Title: Generalized regression neural network association with terahertz spectroscopy for quantitative analysis of benzoic acid additive in wheat flour

Manuscript ID: RSOS-190485

The editor assigned to your paper has now received comments from reviewers. We would like you to revise your paper in accordance with the referee and Subject Editor suggestions which can be found below (not including confidential reports to the Editor). Please note this decision does not guarantee eventual acceptance.

Please submit a copy of your revised paper before 17-May-2019. Please note that the revision deadline will expire at 00.00am on this date. If we do not hear from you within this time then it will be assumed that the paper has been withdrawn. In exceptional circumstances, extensions may be possible if agreed with the Editorial Office in advance. We do not allow multiple rounds of revision so we urge you to make every effort to fully address all of the comments at this stage. If deemed necessary by the Editors, your manuscript will be sent back to one or more of the original reviewers for assessment. If the original reviewers are not available we may invite new reviewers.

Please also include the following statements alongside the other end statements. As we cannot publish your manuscript without these end statements included, if you feel that a given heading is not relevant to your paper, please nevertheless include the heading and explicitly state that it is not relevant to your work.

- Ethics statement

Please clarify whether you received ethical approval from a local ethics committee to carry out your study. If so please include details of this, including the name of the committee that gave consent in a Research Ethics section after your main text. Please also clarify whether you received informed consent for the participants to participate in the study and state this in your Research Ethics section.

OR

Please clarify whether you obtained the necessary licences and approvals from your institutional animal ethics committee before conducting your research. Please provide details of these licences and approvals in an Animal Ethics section after your main text.

OR

Please clarify whether you obtained the appropriate permissions and licences to conduct the fieldwork detailed in your study. Please provide details of these in your methods section.

- Acknowledgements

RSC Associate Editor
Comments to the Author:
The authors must address concerns raised by one of the reviewers.

Reviewers' Comments to Author:
Reviewer: 2

Comments to the Author(s)
The manuscript is substantially improved and the authors have reply to the all comments. I appreciate the work done by the authors to improve the manuscript. The paper is acceptable and no further revision is needed.

Reviewer: 3

Comments to the Author(s)
This paper describes the analysis of benzoic acid (BA) in wheat flour using terahertz spectroscopy (TS) and principal component analysis (PCA) and artificial neural networks (ANN). Although the subject of this paper may be useful, the work here presented is quite insufficient. The following are the main critical points:

- Is the percentage of BA realistic in real applications? Indeed, up to 20% is a huge amount of BA!

Although in the paper, some figures show a concentration of BA up to 0,20%. The maximum concentration was 20 or 0,20%?

- ANN works as a black box that needs optimization. For example, the number of layers, etc. But also, the number of iterations needs to be monitored to avoid over-fitting. What optimization strategy was used?

- The analytical methodology must be validated by the analysis of real samples and the results must be compared with other results obtained by standard methodologies.

Author's Response to Decision Letter for (RSOS-190485.R0)

See Appendix C.

RSOS-190485.R1 (Revision)

Review form: Reviewer 2

Is the manuscript scientifically sound in its present form?

Yes

Are the interpretations and conclusions justified by the results?

Yes

Is the language acceptable?

Yes

Is it clear how to access all supporting data?

Not Applicable

Do you have any ethical concerns with this paper?

No

Have you any concerns about statistical analyses in this paper?

I do not feel qualified to assess the statistics

Recommendation?

Accept as is

Comments to the Author(s)

The authors have sufficiently improved the manuscript. I am very satisfied with the revisions made by the authors. I recommend for publication of this manuscript.

Review form: Reviewer 3

Is the manuscript scientifically sound in its present form?

Yes

Are the interpretations and conclusions justified by the results?

Yes

Is the language acceptable?

Yes

Is it clear how to access all supporting data?

Yes

Do you have any ethical concerns with this paper?

No

Have you any concerns about statistical analyses in this paper?

No

Recommendation?

Accept as is

Comments to the Author(s)

The authors have answered reasonably to the referees suggestions.

Decision letter (RSOS-190485.R1)

25-Jun-2019

Dear Dr Sun:

Title: Generalized regression neural network association with terahertz spectroscopy for quantitative analysis of benzoic acid additive in wheat flour

Manuscript ID: RSOS-190485.R1

It is a pleasure to accept your manuscript in its current form for publication in Royal Society Open Science. The chemistry content of Royal Society Open Science is published in collaboration with the Royal Society of Chemistry. I apologise this has taken longer than usual.

Yours sincerely,

Dr Laura Smith

Publishing Editor, Journals

RSC Associate Editor:
Comments to the Author:
(There are no comments.)

RSC Subject Editor:
Comments to the Author:
(There are no comments.)

Reviewer(s)' Comments to Author:
Reviewer: 2

Comments to the Author(s)
The authors have sufficiently improved the manuscript. I am very satisfied with the revisions made by the authors. I recommend for publication of this manuscript.

Reviewer: 3

Comments to the Author(s)
The authors have answered reasonably to the referees suggestions.

Appendix A

The authors proposed the application of two machines learning models: the GRNN and the standards BPNN. The paper must be significantly improved before to be accepted.

1. Section 2.3 is unclear and must be reformulated.
2. Section 2.4 must be improved. The following papers may be a useful reference and must be reported:

<https://doi.org/10.1109/72.97934>

<https://doi.org/10.1080/10286600500126256>

<https://doi.org/10.1016/j.advengsoft.2005.05.002>

[https://doi.org/10.1061/\(ASCE\)EE.1943-7870.0000435](https://doi.org/10.1061/(ASCE)EE.1943-7870.0000435).

<https://doi.org/10.1007/s00703-012-0205-9>.

<https://doi.org/10.1080/09593330.2013.878396>.

3. The section Results and discussion must be completely reformulated and the results must be presented in a clear manner, separately in the training and in the validation phases.

Appendix B

Comment: 1. “The data available from THz measurements is generally never enough for BPNN or LS-SVM” – Unclear.

Response: The sentence had been deleted.

Comment: 2. References are needed for the statements made on PNN and GRNN in Introduction.

Response: More introduction of GRNN had been added.

Comment: 3. “GRNN can be used for regression, prediction and classification...” – The difference between reg. and pred. is? Moreover, for classification is used GRNN counterpart, i.e. PNN.

Response: This part had been revised. GRNN is used for regression, and PNN is used for classification.

Comment: 4. “...approximately normal around the averaged value of 10.25%.” - Histogram with appropriate test will be useful here.

Response: The histogram had been added.

Comment: 5. “for modelling applicable model.” – You mean creation of?

Response: It has been changed into ‘for developing applicable model.

Comment: 6. “line fitted equation was established for investigating the relationship” – But no discussion is provided on this.

Response: The discussion had been added as the following: According to this principle, further investigation should be executed to mine THz spectra for accuracy improvement. For Lambert Beer’s law, only the fingerprint peak can give a better relationship equation using pretreatment liquid sample. In this case THz spectroscopy can’t obtain a high correlation coefficient because the sample is relatively complex and only fingerprint peak can’t give enough information. Therefore it requires more variables as the input vectors to improve the performance of the models.

Comment: 7. PCA related results must be presented.

Response: The PCA related results had been added.

The number of input vectors will influence the architecture and performance of GRNN model. Generally, the minimal architecture of GRNN is easy to obtain a better generalization of data relation. Principal Component Analysis (PCA) is a bilinear modeling method which gives an interpretable overview of the main information in a multidimensional data table. The information carried by the original variables is projected onto a smaller number of underlying variables called principal components. The first principal component covers as much of the variation in the data as possible. The second principal component is orthogonal to the first and covers as much of the remaining variation as possible, and so on. THz spectrum includes 300 spectral variables in range of 0.5-3THz, and some variables may contain irrelevant information for regression. Therefore, PCA was applied to transform original THz spectra into new axis and obtain the PCA scores as new variables. The scores show the locations of the samples along each model component, and can be used to detect sample patterns, groupings, similarities or differences. The score plots of first and second principal components were shown in Figure 5. The distribution of the calibration samples covered the validation and test ones.

Figure 5. The score plots of first and second principal components.

The variance is computed as the mean square of deviations from the mean. It is equal to the square of the standard deviation. The PCA scores accounted for the greatest amount of variability varied from 97.04% to 99.98% (Figure 6), which presented in the THz spectra collected using THZ-TDS system.

Figure 6. Variance plots for different principal components.

Comment: 8. “The different PCA scores were selected as the input vector of GRNN model for investigating the influence of different input dimensionalities, the results were shown in figure 4.” – You have actually change the number of PCs?

Response: The number of principal components and smooth factor were optimized. The details were as following:

The number of input vectors and smooth factor (σ) of RBF are two importance parameters that influenced the performance of GRNN model. The principal component scores were chosen as the input vectors of GRNN model. The number of principal components varied from one to ten. The smooth factor, representing the spread of RBF, is another important index that affected the performance of GRNN model. The larger the σ , the smoother the function approximation will be. Oppositely, the smaller the σ , the stronger the approximation to the sample will be. In this case,

the method of circle training was adopted to optimize the *spread* in the range of 0.02-2, and the interval was 0.02. The training and validation datasets were used to create GRNN model and optimize the parameters, the results were shown in Figure 7. The optimal GRNN model was obtained with 4 principal components and smooth factor of 0.4.

Figure 7. Optimization for number of principal components and smooth factor.

Comment: 9. Fig 4 data - Is this obtained on test set? If so, this is not good practice, because the credibility of test set is based on principle that it can not be used for model parameter determination. You should use eugenvalues as criterion for PC relevancy.

Response: The datasets had been divided into training, validation and test sets. The training and validation sets were used to create model and optimize parameters. And test set was used to assess the model.

Comment: 10. “The larger the spread, the smoother the function approximation will

be. Oppositely, the smaller the spread, the stronger the approximation to the sample will be. “ - This is not actually the case, and what the stronger approximation means?

Response: The smooth factor behaves as a regularization parameter of RBF. The σ parameter trades off correct prediction of training examples against maximization of the decision function's margin. For larger values of σ , a smaller margin will be accepted if the decision function is better at predicting all training points correctly. A lower σ will encourage a larger margin, therefore a simpler decision function, at the cost of training accuracy.

Comment: 11. “In this case, the method of circle training” – Details are needed.

Response: It has been added.

Comment: 12. In section 3.3. the GRNN results are not presented.

Response: It had been added.

Comment: 13. Fig 5. – What series 3 to 10 means?

Response: The spectra had been handled again, the Figure 5 has been deleted.

Comment: 14. “The best BPNN model was achieved with seven input vectors of PCA scores and six neurons of hidden layer.” - You can not compare models with different inputs!

Response: I have processed the spectra again. The number of input vectors is the same.

Comment: 15. Fig 6. - What dataset was used to determine those meta-parameters?

Response: The validation dataset was used to determine the meta-parameters.

Comment: 16. And finally, GRNN results from figure 7 should be presented in section 3.3. In fig 7 keep only BPNN results. Also, add table with performance metrics for those two models.

Response: It had been added.

Comment: 17. To conclude, the key flaw is that a third dataset is missing. You can not create a reliable ANN without three dataset: training for weights determination, validation for meta-parameters (e.g. number of inputs or spread in GRNN and similar) determination, and test set to assess final model. If you make multiple training runs until you get something that works best on the test data, you have just rendered the test set as training one, and the model you chose as “the best “ will has to be tested once again on “unseen” data, because you've essentially created a ANN model specifically for the test set.

Response: The samples had been divided into training, validation and test datasets. The spectra had been processed again.

Comment: 1. Section 2.3 is unclear and must be reformulated.

Response: This section had been rewritten.

A Fast Fourier Transform (FFT) was adopted to acquire the spectral distribution of the THz pulse in the frequency. The sample's absorption coefficient (α) could be calculated with eq. (1-2).

$$\alpha(\omega) = \frac{2}{d} \ln\left(\frac{4n(\omega)}{p(\omega)(n(\omega)+1)^2}\right) \quad (2.1)$$

$$n(\omega) = 1 + \frac{\phi(\omega)c}{\omega d} \quad (2.2)$$

Where c , ω and d , are the light speed in vacuum, the frequency and the sample's thickness, respectively. The $\rho(\omega)$ and $\Phi(\omega)$ represent respectively the amplitude ratio and phase difference between the reference and sample.

Comment: 2. Section 2.4 must be improved. The following papers may be a useful reference and must be reported.

Response: This section has been rewritten, and the useful references have been cited through the paper.

Figure 1. The architecture of the GRNN.

Differencing from back propagation neural network (BPNN), GRNN is a variation to radial basis neural networks, consists of four layers: input, pattern, summation and output layers [21-23].

Terahertz spectra are used as the input vectors in the first layer. The second layer has the pattern units and the outputs of this layer are passed onto the summation units in the third layer. The final layer covers the output units. Its architecture was shown in Figure. 1. GRNN replaces the sigmoid activation function often used in ANN with a radial basis function (RBF) and achieves the estimation of the probability density function using Parzen's nonparametric estimator [15]. The predicted value is simply a weighted average of the target values of training patterns close to the given input pattern. The smoothing factor σ , representing the width of RBF, is the only adjustable parameter. The details of GRNN are as following:

$$Y(x) = \frac{\sum_{k=1}^N y_k K(x, x_k)}{\sum_{k=1}^N K(x, x_k)} \quad (2.3)$$

$$K(x, x_k) = e^{-\frac{(x-x_k)^T(x-x_k)}{2\sigma^2}} \quad (2.4)$$

Where $Y(x)$ is the prediction value of input x , y_k is the activation weight for the pattern layer neuron at k , and $K(x, x_k)$ is the RBF as formulated above.

Comment: 3. The section Results and discussion must be completely reformulated and the results must be presented in a clear manner, separately in the training and in the validation phases.

Response: This section has been rewritten.

Appendix C

Reviewer: 2

Comments: The manuscript is substantially improved and the authors have reply to the all comments. I appreciate the work done by the authors to improve the manuscript. The paper is acceptable and no further revision is needed.

Response: Thanks

Reviewer: 3

Comments: This paper describes the analysis of benzoic acid (BA) in wheat flour using terahertz spectroscopy (TS) and principal component analysis (PCA) and artificial neural networks (ANN). Although the subject of this paper may be useful, the work here presented is quite insufficient. The following are the main critical points:

Response: We have collected new samples, recorded the spectra and developed model again.

Comments: - Is the percentage of BA realistic in real applications? Indeed, up to 20% is a huge amount of BA! Although in the paper, some figures show a concentration of BA up to 0,20%. The maximum concentration was 20 or 0,20%?

Response: We have collect samples again. The concentrations were from 0.08% to 1.14%. The distribution is close to practical use.

Comments: - ANN works as a black box that needs optimization. For example, the number of layers, etc. But also, the number of iterations needs to be monitored to avoid over-fitting. What optimization strategy was used?

Response: We adapted training, validation and external testing datasets. For avoiding over-fitting, early stop strategy was used as following:

he available data has been divided into the training and validation sets. The former is used for computing the gradient and updating the network weights and biases. The latter is monitored during the training process. The validation error normally decreases during the initial phase of training, as does the training set error. However, when the network begins to over fit the data, the error on the validation set typically begins to rise. When the validation error increases for a specified number of iterations (`net.trainParam.max_fail`), the training is stopped, and the weights

and biases at the minimum of the validation error are returned.

Comments: - The analytical methodology must be validated by the analysis of real samples and the results must be compared with other results obtained by standard methodologies

Response: For testing the model with real samples, we collected ten samples from a local oil and food testing instrument. And these samples were used as external dataset for testing the model performance.

Ethics statement:

We have declared we have no competing interest at the end of the manuscript.